# Exposure to Inorganic Arsenic in Rice in Brazil: A Human Health Risk Assessment

**DOI:** 10.3390/ijerph192416460

**Published:** 2022-12-08

**Authors:** Michele C. Toledo, Janice S. Lee, Bruno L. Batista, Kelly P. K. Olympio, Adelaide C. Nardocci

**Affiliations:** 1School of Public Health, University of São Paulo, São Paulo 01246-904, Brazil; 2United States Environmental Protection Agency, Research Triangle Park, NC 27711, USA; 3Center for Natural and Human Sciences, Federal University of the ABC, Santo André 09210-580, Brazil

**Keywords:** probabilistic risk assessment, hazard quotient, Monte Carlo, dietary exposure risk, mitigation measure, food contaminants

## Abstract

In certain populations, rice is the main source of exposure to inorganic arsenic (iAs), which is associated with cancer and non-cancer effects. Although rice is a staple food in Brazil, there have been few studies about the health risks for the Brazilian population. The objective of this study was to assess the risks of exposure to iAs from white rice and brown rice in Brazil, in terms of the carcinogenic and non-carcinogenic effects, and to propose measures to mitigate those risks. The incremental lifetime cancer risk (ILCR) and hazard quotient (HQ) were calculated in a probabilistic framework. The mean ILCR was 1.5 × 10^−4^ for white rice and 6.0 × 10^−6^ for brown rice. The HQ for white and brown rice was under 1. The ILCR for white and brown rice was high, even though the iAs concentration in rice is below the maximum contaminant level. The risk for brown rice consumption was lower, which was not expected. Various mitigation measures discussed in this report are estimated to reduce the risk from rice consumption by 5–67%. With the support of public policies, measures to reduce these risks for the Brazilian population would have a positive impact on public health.

## 1. Introduction

Although arsenic exists in various chemical forms, it is mainly categorized, from a public health perspective, as organic or inorganic. These different forms occur naturally in the environment, and anthropogenic activities can substantially increase their concentration and bioavailability in soil and water, allowing them to be absorbed by plants in agricultural fields. Inorganic arsenic (iAs) is classified as a Group 1 carcinogen and is present in trace amounts in rocks, soil, air, food, and water [1,2]. Food and water are the most important sources of exposure to iAs, and rice (*Oryza sativa*) is the main source of exposure in some populations [3,4]. Rice is usually cultivated in flooded fields, where the anaerobic conditions increase iAs availability in soils. Arsenate is reduced to the more mobile arsenite, leading to a higher concentration of both forms close to the plant roots. Arsenate and arsenite are analogues of the plant micronutrients phosphate and silicic acid, respectively, therefore being easily taken up and stored by the plant. Fertilizers, pesticides, and the water used to irrigate the crops can also be sources of iAs [5,6].

Rice is a staple food for more than half of the world population, accounting for approximately 30% of the energy intake and 20% of the protein intake; it can, therefore, be a significant source of iAs and other metals [7,8,9]. The European Food Safety Authority has recognized rice as the main source of iAs for the European population [10]. In Asia, rice is also a staple food and tends to be the major source of iAs from food [11]. In a study conducted in China [12], the concentrations of iAs and other metals in rice were found to be above the maximum contaminant levels (MCLs) of 200 ng g^−1^ established by the United Nations Food and Agriculture Organization and the World Health Organization (FAO/WHO).

In Brazil, rice is one of the main components of daily meals and is consumed by the entire population, regardless of socioeconomic status. According to a national study conducted by the Brazilian Institute of Geography and Statistics, the average daily consumption of rice by adults is estimated to be 167 g [13], which is comparable to that reported for some locations in China [14]. Rice consumption accounts for 46–79% of the iAs ingested by the Brazilian population [15].

Epidemiological studies have shown that exposure to iAs is associated with cancer of the lung, bladder, kidney, skin, liver, and prostate [2]. The non-cancer effects from long-term oral exposure to iAs include dermal, cardiovascular, respiratory, and neurodevelopmental changes, and acute high-dose oral exposure has been associated with nausea, vomiting, diarrhea, and encephalopathy [16]. The susceptible life stages are pregnancy, infancy, and early childhood [4]. The Joint FAO/WHO Expert Committee on Food Additives recognizes iAs in rice as a public health concern [17], and quantitative risk assessment has been considered an important tool for risk management and to support decision-making in public health [18].

Only a few studies have evaluated exposure to iAs in rice in Brazil [15,19,20,21,22,23,24]. In the present study, we focused on a probabilistic analysis of the risk of exposure to iAs in polished (white) and husked (brown) rice from Brazil. Hypothetical scenarios of risk reduction were also assessed. Cancer and non-cancer risks were estimated using the slope factor and the U.S. Environmental Protection Agency (EPA) reference dose for iAs [25].

## 2. Materials and Methods

### 2.1. Average Daily Dose, Cancer Risk, and Hazard Quotient

The average daily dose by exposure pathway, which can account for differences between the age group of the exposed people regarding exposure factors such as body weight and eating habits, was estimated according to the following equation [26]:(1)$$ADD_{j}=\frac{\left[C\times IR_{j}\times ED_{j}\times EF_{j}\right]}{\left[BW_{j}\times AT\right]}\;\;$$
where *ADD_j_* is the average daily dose in mg/kg-day estimated for the age group *j*; *C* is the concentration of iAs in raw rice (mg iAs/g rice); *IR_j_* is the ingestion rate of rice (g rice/day) for the age group *j*; *ED_j_* and *EF_j_* are the exposure duration (years) and exposure frequency (days/year) for the age group *j*, respectively; *BW_j_* is the body weight for the age group *j*; and *AT* is the average time, which is the *ED_j_* × 365 days.

As the exposure varies with age, the incremental lifetime cancer risk (*ILCR*) was estimated by summing the cancer risk in each age group [27], as follows:(2)$$ILCR=\overset{\;}{\underset{j=1,n}{\sum }}(ADDj\times SF)\times \frac{ED_{j}}{LT})\;$$
where *SF* is the slope factor for oral iAs, which is 1.5 (mg/kg-day)^−1^, *LT* is the lifetime, which is 70 years, and *n* is the number of age intervals. The *ILCR* estimates the incremental lifetime cancer risk for exposure to certain carcinogens. The result must be interpreted as a probability, represented as a value between 0 and 1. A risk of 10^−5^ indicates a probability of 1 chance in 100,000 of an individual developing cancer [26].

The risk of non-cancer effects (cardiovascular and dermal outcomes) was estimated by the hazard quotient (*HQ*) summing the fractional *HQ* of each age group, which is the *ADDj* divided by the *RfD* weighted by exposure duration *j*:(3)$$HQ=\overset{\;}{\underset{j=1,n\;}{\sum }}\left(\frac{ADD_{j}}{RfD}\right)\;\times \frac{ED_{j}}{LT}\;$$
where *RfD* is the reference dose for oral exposure to iAs, which is 0.003 mg/kg per day [25], and *n* is the number of age intervals. An *HQ* above 1 indicates that the average daily dose is higher than the reference dose, so there may be concern for potential noncancer effects.

The *ILCR* and *HQ* calculations (Appendix A) were implemented in a probabilistic framework, with a Monte Carlo simulation of 100,000 iterations and with a confidence interval of 95%. The simulations were performed using distributions for the concentration of arsenic in rice. The probabilistic assessment was carried out in the open-source software YASAIw from the State of Washington Department of Ecology [28]. A limitation of this method is that the probabilistic analysis is an indirect measure of the risk.

### 2.2. Concentration of iAs in Brazilian Rice

The data related to the concentrations of iAs in white and brown rice (Appendix A) were obtained from a study conducted by Batista et al. [29], one of the few studies about iAs in rice from Brazil to which we had access. In that study, samples of raw rice were collected from different areas in Brazil, mainly the southern region, which is the largest rice-producing region in the country. There were 64 samples of white rice (all obtained from markets) and 90 samples of brown rice (69 from farms and 21 from markets). Firstly, the authors determined the total concentration of arsenic by microwave digestion of the rice samples, as described by Paniz et al. [30]. For this purpose, the samples were ground, sieved (<250 µm), and weighed (200 mg) in triplicate and then placed in 100-mL polytetrafluoroethylene vessels, where 4 mL of sub-distilled HNO_3_ (20 vol%) and 1 mL of H_2_O_2_ (30 vol%) were added. The tubes were then placed in a microwave system (up to 35 bar). After cooling, ultrapure water was added to make up 50 mL, and then the samples were analyzed by inductively coupled plasma mass spectrometry (ICP-MS). Secondly, the concentration of arsenic chemical species was determined. The speciation analysis was performed as described by Batista et al. [19]. The samples, in two replicates, containing about 200-mg of ground and sieved (<250 µm) rice, were weighed into 50-mL conical tubes, treated with 10 mL of HNO_3_ (2 vol%), and then stirred (100 rpm) for 24 h. In sequence, the tubes were heated (95 °C) in a water bath for 2.5 h. Finally, after cooling, the samples were analyzed using a high-performance liquid chromatography coupled to an ICP-MS. The concentration of iAs was considered the sum of the species arsenite and arsenate, and the concentration of organic arsenic was considered the sum of the species dimethyl arsenic and monomethyl arsenic. In this risk assessment, we only included iAs in the analysis.

### 2.3. Rice Consumption

The rice consumption for each age group, which includes only rice grain consumption (not rice-derived products, such as rice flour, present in certain kinds of food), was obtained from different sources, as also presented in Appendix A (Appendix A):From 4 months to 1 year and 5 to 10 years—the estimated rice consumption was based on the quantity recommended by the São Paulo Municipal Department of Education for consumption in schools and daycare centers [31,32]. Since we noticed that the recommended consumption was overestimated in comparison with the actual consumption for children 1 to 5 years of age (see below), we estimated that the actual rice consumption for children 4 months to 1 year and 5 to 10 years of age was 39.8% less than that. In the school diet, rice is served cooked (as a side dish, in soups, or as a dessert) and the quantities were registered in grams of raw rice. Students from 6 to 10 years of age have a part-time school period, having only one meal at school (lunch or dinner), and daily rice consumption for that age group was, therefore, estimated on the basis of the recommended quantity.From 1 to 5 years—data were obtained from a study involving 64 children at 2 daycare centers in the city of São Paulo [33], in which the portions of 24-h duplicate diet samples were recorded, including the food consumed at the daycare center and that consumed at home. The consumption of rice and soup containing rice was evaluated. Household measures (e.g., tablespoons) were converted to grams in accordance with nutrition guidelines [34,35].From 10 to 70 years—data were obtained from a study conducted by the Brazilian Institute of Geography and Statistics [13], in which the consumption of cooked rice in the last 48 h was determined on the basis of self-reports by interviewees in all Brazilian states, from 10 to 70 years.

Since there are no available data regarding daily consumption of brown rice in Brazil, the same rates were considered for white and brown rice.

### 2.4. Body Weight

Body weight was also obtained from the study conducted by the Brazilian Institute of Geography and Statistics [36]. Body weight was calculated by the weighted average of the male and female population, in relation to the total of interviewed individuals.

### 2.5. Exposure Frequency

Since rice is a staple food in Brazil, the exposure frequency (EF) for rice was considered to be 6 days/week (312.85 days/year). 

### 2.6. Statistical Analysis

The statistical analysis was conducted with R software, version 3.5.0, and R studio, version 1.1.453 (The R Foundation for Statistical Computing, Vienna, Austria). We used the fitdistrplus package in order to fit the distribution of the iAs concentration datasets. The 64 white rice and 90 brown rice iAs concentrations from Batista et al. [29] were fit using the normal, lognormal and exponential distribution assumptions available in the fitdistrplus package. The distribution with the lowest Akaike information criterion value was selected. Pearson’s correlation coefficient was calculated to identify a correlation between iAs and cadmium in rice as evidence of water management during rice cultivation.

Appendix A (Appendix A) summarizes the exposure parameters of body weight and rice dietary consumption rates for various age groups, obtained or estimated from the sources described above and adopted for the risk assessment. 

The data for rice consumption and body weight did not have a good fit in any distribution, so to minimize the uncertainty, avoiding using mean values for the lifetime, the risk was assessed for age groups. The advantage of this approach is to obtain risk results for each age group, with it being possible to identify which is more vulnerable.

### 2.7. Hypothetical Scenarios of Risk Reduction

Adopting the same dose–response model for each of six different scenarios, we assessed hypothetical interventions intended to reduce the risk (Appendix A). The objective was to simulate the impact of different mitigation strategies, some of which were proposed by the U.S. Food and Drug Administration (FDA)—such as lowering the MCL or interrupting the exposure of infants and children)—whereas others were based on our results.

## 3. Results

Figure 1 shows a box plot of iAs concentrations in white and brown rice. Appendix A shows the descriptive statistics (Appendix A). The normal distribution had the best fit for the white rice dataset (mean, 100.2 ± 44.6), whereas the log-normal distribution had the best fit for the brown rice data (μ = 4.1; δ = 0.9).

### 3.1. Cancer Risk Associated with Exposure to iAs in Rice

Table 1 presents the results of cancer risk by age and ILCR, and Appendix A (Appendix A) presents the ADD by age. The mean and 95th percentiles of the total incremental lifetime cancer risk (ILCR) were 1.51 × 10^−4^ and 2.60 × 10^−4^ for white rice and 6.04 × 10^−6^ and 8.47 × 10^−6^ for brown rice. A limitation of this method is that it is not possible to determine the risk of a specific cancer, but the overall cancer related to exposure to iAs (liver, kidney, lung, bladder, and skin).

### 3.2. Non-Cancer Risk of Exposure to iAs in Rice 

The results of our analysis of the non-cancer risks of exposure to iAs in white and brown rice in Brazil, estimated by calculating the fractional HQ by age and the lifetime HQ, are presented in Table 2. The fractional and lifetime HQ for white and brown rice consumption were below 1 for individuals of all ages. The lifetime HQ found is 3.37 × 10^−1^ for white rice and 1.34 × 10^−2^ for brown rice. An HQ above 1 indicates that the dose ingested is higher than the reference dose and that there is a potential for adverse effects, in this context including cardiovascular and dermal effects [37].

### 3.3. Exposure to iAs in Rice

In the present study, the mean concentration of iAs in white rice was found to be 100.1 ± 44.62 ng g^−1^, which is lower than the MCL of 300 ng g^−1^ proposed by the BNMH for total arsenic [38]. That value is also lower than the MCL of 200 ng g^−1^ for iAs established by the Joint FAO/WHO Codex Committee on Contaminants in Foods [39], European Commission [40], and Chinese Ministry of Health [41]. Among the 64 samples of white rice evaluated in the present study, the concentration of iAs was above the FAO/WHO MCL (ranging from 200 ng g^−1^ to 220 ng g^−1^, and thus in accordance with Brazilian MCL) in only 4 samples. 

The mean concentration of iAs in brown rice was found to be 80.1 ± 55.5 ng g^−1^. In Brazil, the MCL is the same (300.0 ng g^−1^) for brown and white rice, although the Joint FAO/WHO Codex Committee on Contaminants in Foods recommends an MCL of 350.0 ng g^−1^ for brown rice [42]. 

Many factors that can affect the concentration of iAs in rice can be associated with the location of the rice field [43]. Therefore, we conducted an analysis based on the location where the samples were collected, although we have that information only for the samples of brown rice obtained directly from farms. The samples of white rice were all obtained from markets, and the specific cultivation location was not noted on any of the labels [29]

Figure 2 shows a box plot of iAs concentrations in brown rice by cities where the farms were located [29]. Only cities with more than two samples were included, resulting in three of eight cities. The mean iAs concentration in samples from City 1 was 64.0 ± 19.0 ng g^−1^ (*n* = 9), 45.1 ± 43.0 ng g^−1^ (*n* = 30) from City 2, and 79.8 ± 22.0 ng g^−1^ (*n* = 14) from City 3. City 2 presented the highest variation, but on average, the three locations produced rice with concentration of iAs under 100 ng g^−1^, suggesting that the studied farms produced rice with a low concentration of iAs. 

Figure 3 shows a box plot of iAs concentrations in brown rice by the origin of the samples (farms or markets), which were similarly processed [29]. The mean concentration of iAs for the samples obtained from markets (*n* = 21) of 154.91 ± 44.8 ng g^−1^ (range, 135.7–222.8 ng g^−1^) was similar to that reported in the literature, but much higher than the 57.36 ± 34.6 ng g^−1^ (range, 45.1–79.7 ng g^−1^) estimate from the samples obtained from farms (*n* = 69). That discrepancy influenced the overall mean iAs concentration in the dataset for brown rice. The farms where the brown rice was cultivated are located in two states in the southern region of Brazil—Rio Grande do Sul and Santa Catarina—the two main rice-producing states in the country, accounting for approximately 69% and 9% of the national rice production, respectively [44]. Of the brown rice samples purchased in markets, most were produced in Rio Grande do Sul or São Paulo, although the labels did not identify the specific cities, and 12 of the samples had labels that provided no information regarding the state in which the rice was grown. Possible explanations for the lower iAs concentrations in brown rice obtained from farms include the location and management of the farms, which receive support from the Brazilian Agency for Agricultural Research, a governmental agency linked to the Ministry of Agriculture, Livestock, and Food Supply. The mission of the agency is to improve agriculture practices, and one of its goals is to achieve food safety and food security, providing support for farmers to produce more food and food free of hazardous substances [45]. Adopting good agriculture practices, such as avoiding contamination sources and implementing water management, as well as monitoring soil and water quality, could indirectly result in lower iAs concentrations in rice.

In the southern region of Brazil, rice is usually cultivated in flooded fields. However, we hypothesized that rice farmers could be cultivating rice in unsaturated soils, which could explain the low concentration of iAs in rice, although it would also result in higher cadmium concentrations. Using cadmium concentration as an indicator, we found no linear correlation between arsenic and cadmium concentration in brown rice from farms (r = 0.049; *p* = 0.690), so there is no evidence that the rice was grown in unsaturated soils. Therefore, the low concentration of iAs is probably attributable to other factors.

### 3.4. Reducing the Risk 

The concentration of arsenic in rice grains depends on the arsenic concentration in soil, its bioavailability, and the rice genotype (cultivar). Since some rice cultivars reportedly store less arsenic, selecting those cultivars could be a good strategy when the soil is known to contain bioavailable arsenic [46]. 

In the present study, we had information about brown rice varieties only for the samples collected directly from farms (i.e., not for those obtained from markets). Although there are four rice varieties, we had more than one sample for only two. The mean iAs concentration was 80.7 ± 22.35 ng g^−1^ for the Irga 424 variety (*n* = 19) and 46.6 ± 31.69 ng g^−1^ for the Puitá variety (*n* = 48), as shown in Figure 4. Adopting this rice variety could be an easy, effective strategy to reduce health risks for the population.

The results of the present risk assessment describe the estimates of the incidence of cancer in the Brazilian population, based on the application of EPA’s cancer slope and exposure scenarios adopted. By changing the inputs of the Monte Carlo analysis, we can simulate the likely impact of mitigation strategies [4]. We evaluated six different interventions aimed at reducing the cancer risk, calculating the risk for each of those interventions: scenario 1—consumption exclusively of brown rice from farms with low levels of iAs; scenario 2—consumption exclusively of brown rice of the Puitá variety, which was found to have the lowest iAs concentration (mean, 46.6 ng g^−1^); scenario 3—adoption of a white rice MCL of 100 ng g^−1^; scenario 4—adoption of a white rice MCL of 75 ng g^−1^; scenario 5—adoption of a white rice MCL of 50 ng g^−1^; and scenario 6—no consumption of white rice by infants and children ≤ 6 years of age (we chose white rice because it is the type of rice most widely consumed in Brazil). The U.S. FDA has proposed interventions similar to those evaluated here [4]. To calculate the risk for those hypothetical scenarios, any samples above the proposed limit were removed. For scenarios 1 and 2, the complete dataset was considered and the iAs limit was the maximum concentration found. Table 3 shows the parameters and the results of the risk assessment for each scenario. 

There is no guideline establishing an acceptable level of risk associated with exposure to arsenic in food. Some studies have used the EPA guideline for contaminated areas, which established an acceptable cancer risk ranging from 10^−4^ to 10^−6^ [12,47,48]. Scenario 2 presented the lowest ILCR, close to 10^−6^, in which the MCL was 48 ng g^−1^, more than 7 times lower than the MCL proposed by the FAO for white rice [39]. In this scenario, there is a reduction of almost 11% of the risk, compared with the risk of consuming brown rice. It is likely that polishing Puitá rice would further reduce the iAs content and, consequently, the risk.

Scenario 5 represents the adoption of an MCL of 50 ng g^−1^ for white rice, the kind of rice most widely consumed in Brazil, resulting in the highest decrease in ILCR, around 68% (compared with the risk from consumption of white rice) and third lowest ILCR. Excluding rice from the diet of infants and young children (scenario 6) is also a scenario proposed by the FDA [4], and it could reduce the ILCR by nearly 23%, although it would necessitate a substantial change in Brazilian culture, which is unlikely to happen. Daycare centers introduce rice into the diet of infants at four months of age, and that could be delayed, with another type of food—also rich in nutrients and unprocessed—being prioritized. The ILCR for scenario 6 is similar to that of reducing the MCL to 100 ng g^−1^ (scenario 3). In this context, Segura et al. [49] emphasized the need for crop-tracking, given that the iAs content in rice can vary significantly, even among samples from the same producer. That would allow the selection of grains with less iAs for consumption by vulnerable populations, such as infants and children.

## 4. Discussion

Rice consumption starts at an early age in Brazil. According to food consumption guidelines for daycare centers in the city of São Paulo, the consumption of rice and other kinds of solid food starts at 4 months of age [31]. Since the standard maternity leave in Brazil is 120 days [50], exclusive breastfeeding until 6 months of age, as recommended by the WHO [51], is a challenge. Measures to increase maternity leave in the country have recently been proposed. Since 2008, civil servants have had 180 days of maternity leave, and there are tax incentives for companies that grant 180 days of maternity leave to their employees [52]. However, the proportion of the workforce protected by labor laws that guarantee maternity leave has decreased [53].

The ILCR for exposure to iAs in white rice obtained in our study is lower than that reported for other countries where rice is also a staple food. In Saudi Arabia, Al-Saleh and Abduljabbar [12] found a mean ILCR of 5.8 × 10^−2^ (minimum of 1.2 × 10^−2^ and maximum of 2.6 × 10^−1^). The authors used the concentrations of total arsenic but estimated that iAs represented 80–90% of the total. Another probabilistic risk assessment, conducted in China by Li et al. [14], found an average ILCR of 1.77 × 10^−3^, higher than that found in the present study. Those authors evaluated the consumption of white rice and other foods, such as rice flour, coarse cereals, vegetables, fruit, meat, milk, eggs, and aquatic products. They found that the most relevant variable was the rate of ingestion of aquatic products, followed by the iAs concentration in rice. The ILCR varied among the different areas of the country, and the authors concluded that the risk was mainly explained by the kind of food consumed and the ingestion rate. 

In Taiwan, Chen et al. [54] found the mean ILCR for exposure to iAs in white and brown rice to be 1.04 × 10^−4^ for males and 7.87 × 10^−5^ for females. The mean ILCR for exposure to iAs in white rice was reported to be 2.06 × 10^−4^ in Punjab, India [47]. These results are similar to the result found in our study.

A major risk assessment conducted in the United States by the FDA found a median ILCR of 3.4 × 10^−5^ for white rice (with 5% and 95% confidence limits of 0 and 6.9 × 10^–5^, respectively) and 5.4 × 10^−6^ for brown rice (0 and 1.1 × 10^–5^ are the confidence limits of 5% and 95%, respectively) [4], both of which are lower than the values found in our study. That could be attributed to the fact that rice consumption is higher in Brazil. The authors of that study also calculated the risk associated with a higher—but still lower than that reported for Brazil—per serving (per eating occasion) dose level. On that basis, the median risk would be 1.36 × 10^−4^ for white rice (0 and 2.78 × 10^−4^ are the 5% and 95% confidence limits, respectively) and 1.64 × 10^−4^ for brown rice (0 and 3.38 × 10^−4^ are the 5% and 95% confidence limits, respectively).

The high ILCR values found in the present study are mainly associated with the elevated rice consumption in Brazil, which is on average for all age groups 156.6 g/day, compared with 17.1 g/day (including rice flour) in the United States [4]. According to Meharg [50], rice consumption is also very low (10.0 g/day) in the United Kingdom. However, rice consumption is much higher in most Asian countries, such as China, where the daily rice consumption can be as high as 218.64 g [55], as well as Bangladesh, Laos, and Myanmar, where it ranges from 400.0 g/day to 500.0 g/day [56].

Rice and beans make up 25% of the diet of the Brazilian population [13]. In Brazil, there are over ten popular dishes and desserts prepared with rice. The consumption of rice and beans is considered healthy compared with that of ultra-processed food, which are formulations of ingredients created by a series of industrial techniques and processes, such as packaged snacks, pre-prepared meat, pasta and pizza dishes, and others [57]. Rice and beans are rich in nutrients and calories, and their consumption can guarantee the daily ingestion of 50% of the recommended daily water intake. In Brazil, rice is also less expensive than is ultra-processed food and is accessible for people of all socioeconomic levels [58]. Approximately 83% of the Brazilian population consumes white rice, and about 4% consumes brown rice [13]. Given that the current population of Brazil is approximately 211 million, the population exposed to iAs in white and brown rice could be approximately 175 million and 8.4 million people, respectively.

The risk of consuming white rice is higher than is that of consuming brown rice, which is attributed to the lower concentration of iAs in brown rice. The Dietary Guidelines for the Brazilian Population established by the BNMH recommend the consumption of brown rice, rather than white rice, because of the composition, in terms of micronutrients and dietary fiber, of the former [58]. Jo and Todorov [59] reported that, as a result of the polishing process, white rice contains lower concentrations of phosphorus, potassium, manganese, and iron than does brown rice. Considering the results of the present risk assessment, we could conclude that brown rice is also a better option for human health. However, the sample size in this study does not allow for the conclusion that all Brazilian brown rice has less iAs than white rice. Further investigations of iAs in Brazilian rice are needed, given that the levels of arsenic in rice vary according to soil properties, type of irrigation, plant characteristics, and other factors. Samples of white and brown rice from the same location would be more appropriated for a comparative analysis.

Brown rice is typically reported to contain higher levels of arsenic than does white rice, because the arsenic is mainly stored in the external layers of the grain, which are partially removed when the grain is polished [60,61]. However, in the present study, the iAs concentration was found to be slightly lower in brown rice than in white rice, as confirmed by a hypothesis test (*t*-test, 95% confidence interval). The samples of brown and white rice came from different locations, and iAs concentration in rice can vary considerably according to the region of origin [43].

Regarding non-cancer risk, some studies have found HQ values above 1 for exposure to iAs in rice. In a study involving adults in Saudi Arabia, Al-Saleh et al. [12] found an HQ of 1.2 (SD = 0.4) for exposure to iAs in white rice. The authors assumed a daily rice consumption of 160 g/day to calculate the dose, and used the reference dose established by the EPA, as was used in the present study. In India, where arsenic in rice is a public health problem in some regions, Upadhyay et al. [62] estimated an HQ above 1 for all age groups and a correlation test suggested that the risk of arsenic poisoning is higher among infants and children than among adults. In a study conducted in Taiwan, Chen et al. [54] found an HQ below 1 for all individuals ≤ 65 years of age (between 0.08 and 0.3), although they adopted a 5-fold higher reference dose of 0.015 mg/kg per day. The same RfD was adopted by Lin et al. [63], which conducted a study of the non-cancer risk of exposure to total arsenic in rice for adults in 14 cities in China and found HQ values below 1 for all of the cities, between 0.07 and 0.3. In Punjab, India, Sharma et al. [47] found an HQ of 0.45 for exposure to total arsenic in specific rice varieties and an exposure period of 70 years, using the RfD proposed by EPA.

The risk results are affected by the concentration of iAs in rice, and similar concentrations have been found in other countries, including the United States and China. According to the FDA [4], the weighted mean concentration of iAs in 429 samples of different types of white rice based on the relative market-share estimates in the United States is 92.3 ng g^−1^ (standard error, 1.3). Considering different grain sizes, the mean concentration of iAs was 102.0 ng g^−1^ for long grain rice (*n* = 173), 81.5 ng g^−1^ for medium grain rice (*n* = 94), and 78.9 ng g^−1^ for short grain rice (*n* = 23). Similarly, in China, where rice is a staple food, Li et al. [14] reported an iAs concentration of 103.0 ng g^−1^ in 151 samples of white rice from published studies. Nevertheless, higher iAs concentrations, ranging from 290.0 ng g^−1^ to 950.0 ng g^−1^, exceeding the WHO MCL and more than 9 times higher than the mean concentration found in our study, were found in white rice samples from India [62]. Lower levels were found in white rice from Taiwan, where rice is the primary staple food, the mean iAs concentration being 65.9 ng g^−1^ in 51 samples [54], and from Iran, a rice-producing country, the mean iAs concentration in 15 samples being 82.0 ng g^−1^ [64].

In Brazil, only a few studies have evaluated the concentration of iAs in white rice. One study assessed white rice purchased in local markets in the state of Minas Gerais, in the southeastern region of the country, and found an iAs concentration of 102.0 ng g^−1^ [15], comparable to that observed in the present study. Cerveira et al. [65] reported a mean iAs concentration of 94.2 ± 39.5 ng g^−1^ (range, 54.0–150.0 ng g^−1^; *n* = 7), similar to our findings. Moreover, in the state of Minas Gerais, Corguinha et al. [66] found low concentrations of total arsenic, below the detection limit of 15.0 ng g^−1^, which was attributed to a low concentration of arsenic in soil. In the present study, we have access to data of three samples from Minas Gerais and found iAs concentrations ranging from 105.0 ng g^−1^ to 132.0 ng g^−1^, higher than the values reported in either of the studies cited above [29].

Higher mean concentrations of iAs were found in brown rice from Taiwan [54]: 103.9 ± 45.0 ng g^−1^ for arsenate and 2.2 ± 1.2 ng g^−1^ for mobile arsenite (*n* = 13). The U.S. FDA [4] reported a mean iAs concentration in brown rice of 156.5 ng g^−1^ (range, 34.0–249.0 ng g^−1^), in 120 samples, as well as reporting a mean iAs concentration in 144 samples of jasmine, basmati, parboiled, and pre-cooked brown rice of 153.8 ± 3.2 ng g^−1^. In contrast, Fu et al. [67] reported a mean predicted concentration of iAs in 282 samples of brown rice from Hainan, an island in China, of 57.0 ng g^−1^, even lower than the concentration found in our study. That concentration was considered lower than or similar to that reported for other regions of China, which the authors suggested was attributed to soil properties (organic matter, phosphorus content, humic acid, and iron–manganese) and arsenic speciation in soil.

Other studies conducted in Brazil have reported concentrations of iAs in brown rice higher than those adopted in the present study, reported by Batista et al. [24]. Cerveira et al. [65] found concentrations ranging from 88.0 ng g^−1^ to 163.0 ng g^−1^ (*n* = 4), with a mean value of 131.0 ± 32.0 ng g^−1^. Batista et al. [19] reported a mean iAs concentration of 188 ng g^−1^ (range, 176.0–202.0 ng g^−1^) in samples of brown rice from the states of Rio Grande do Sul and São Paulo. Kato et al. [23] found significant variation in the levels of total arsenic in brown rice from the states of Rio Grande do Sul (235.0 ± 157.0 ng g^−1^), Santa Catarina (157.0 ± 108.0 ng g^−1^), and Mato Grosso (4.0 ± 2.0 ng g^−1^), which was attributed to differences in water management and local features.

The concentration of iAs in rice can vary according to the presence of arsenic in soil or in water used for irrigation, natural or otherwise [68]; the current or past use of pesticides containing arsenic; anthropogenic sources of iAs, such as mining or industrial activities, near the rice paddies [5]; water management [69]; and rice variety [70,71]. In Brazil, some studies have identified significant variation in iAs concentrations and other non-essential elements in rice, even among rice grains from the same producer. Rice variety, microclimatic conditions, and geochemical properties are reported to be major factors affecting iAs concentration in rice [49,72].

Water management can influence arsenic concentration in rice. Rice cultivated under flooded conditions absorbs more arsenic than does that cultivated in unsaturated soils, while also absorbing less cadmium from soil [69]. Silva et al. [70] evaluated the iAs concentration in three different varieties of rice cultivated under different water conditions and during different phases of development. They found that water management had the greatest impact on iAs concentration during the reproductive period, in which cultivation in unsaturated soils resulted in the lowest arsenic accumulation in rice grains, as well as the highest accumulation of cadmium and lead. Both were below the acceptable levels established by the FAO/WHO [39].

Some types of fertilizers, pesticides, and soil acidity correctors, such as limestone, can be a source of arsenic in the environment. In another study conducted in Brazil, Avelar et al. [73] analyzed samples of limestone, a natural unprocessed mineral, and found an arsenic concentration of 11.74 ± 1.42 µg g^−1^, similar to values reported for limestone in the United States, where the U.S. FDA has declared it a major source of arsenic in the soil, posing risks for humans and animals [4]. The anaerobic conditions in flooded fields favor pH correction, so no limestone is necessary. However, in some cases, rice seeds are sown directly onto dry soil, and the field is flooded 30 days later. Therefore, the recommendation is to use limestone only once every five years [74].

Phosphorus fertilizers can also be a source of arsenic in the environment. Avelar et al. [73] evaluated samples of phosphorus fertilizers in Brazil and found concentrations of total arsenic similar to or even lower than those reported for other countries around the world—11.74 ± 1.42 µg g^−1^. Those concentrations were below the MCL established for Brazil, although Brazilian soils demand more phosphorus-rich fertilizers, because iron and aluminum are more likely to be adsorbed by soil particles than is phosphorus. In comparison with Europe, Brazil uses nearly 100 times more phosphorus fertilizer in agricultural fields (140 kg/ha). On average, 6.4 ± 1.2 g/ha of arsenic is added to the soil in Brazil every year. Although the arsenic concentrations in fertilizers do not pose a threat to human health in the short term, intensive medium- to long-term use of such fertilizers could lead to the accumulation of arsenic in soil, which can represent risks to human health, and soil monitoring is therefore necessary [75]. In a study conducted in the state of São Paulo, Campos [76] found that the intense use of phosphorus fertilizers for decades increased the soil concentrations of arsenic, as well as its mobility and availability, given that phosphorus, rather than arsenic, can be adsorbed by the soil. That increased the concentration of arsenic in groundwater and, consequently, in well water. In Rio Grande do Sul, the main rice-producing state in Brazil, contamination of soil and groundwater with arsenic, due to fertilizer factory activities, has been reported. In a study conducted in the Patos Lagoon Estuary, which is surrounded by rice paddies, Mirlean and Roisenberg [77] reported arsenic concentrations ranging from 7.5 µg g^−1^ to 27.5 µg g^−1^ in soil, exceeding the local background value (1.02 µg g^−1^), and from 1.23 µg g^−1^ to 25.45 ng mL^−1^ in water, also exceeding the local background value (0.14 ng mL^−1^). The authors concluded that the soil and water contamination are a result of precipitation from factory emissions, over a period of more than 40 years, accumulating total arsenic in the superficial horizon of the soil. Most studies of arsenic in the environment of Brazil, including those conducted in the states of Espírito Santo, Bahia, Rio de Janeiro, Paraná, and São Paulo [78,79,80,81], have attributed it to industrial or mining activities, which do not typically occur near rice paddies.

The rice cultivar is also an important factor for arsenic concentration in rice, as shown in Figure 3. In a study conducted in Punjab, India, Sharma et al. [47] investigated two rice varieties (PUSA1121 and PR122) and found that they may be suitable for cultivation in fields contaminated with arsenic. Another study conducted in India showed significant variability of arsenic concentration in five rice varieties [62]. In that study, the Ranjit variety showed a mean iAs concentration of 290 ± 0.021 ng g^−1^, more than three times lower than the 950 ± 0.044 ng g^−1^ shown by the Gosai variety.

A number of measures for reducing the risk of iAs exposure associated with rice consumption have been proposed. Practices that can be adopted by the consumers, including rinsing, soaking, and some cooking methods, can remove part of the iAs. However, there are uncertainties regarding the effectiveness of such practices and the influence of the type of rice; in addition, the iAs content in the water used in cooking the rice can affect the final iAs concentration [4,82].

Sengupta et al. [83] reported that a method of washing and cooking rice, specific to India, removed up to 57% of the total arsenic from rice containing 203–540 ng g^−1^ of arsenic. Washing the rice approximately six times, until the water is clear, reduced the total arsenic concentration by approximately half, and cooking the rice at a rice-to-water ratio of 1:6, thereafter discarding the excess water, removed the remaining arsenic. A study conducted in Saudi Arabia showed that soaking rice for 20 min removed 98% of the total arsenic and that rinsing rice 3 times removed 97% [12]. In a study conducted in Japan, Naito et al. [84] observed that rinsing rice removed mainly iAs. Although rinsing and cooking practices can reduce the arsenic concentration in rice, such practices also reduce enriched iron, folate, thiamin, and niacin [4]. One study showed that rinsing rice before cooking had a minimal effect on arsenic concentration, while removing nutrients such as enriched iron, folate, thiamin, and niacin. Cooking rice in excess water proved to be more effective in reducing the iAs concentration in rice, removing 40–60% depending on the type of rice, while reducing those same nutrients by 50–70% [85]. Cooking the rice can also change the speciation of the arsenic to a form that is more toxic or less toxic, depending on the type of rice and its region of origin [43]. Preliminary estimates indicate that the reduction in iAs in rice from rinsing and cooking practices in water containing low levels of arsenic (<3 µg/L) ranges from 28% to 60%. Since there is substantial uncertainty in those estimates, there have been calls for further research to evaluate not only changes in total arsenic and iAs concentrations in rice but also the impact on nutritional content [4].

Nachman et al. [86] proposed actions that stakeholders (regulators, food producers, researchers, and health professionals) could take at each step of the supply chain to reduce the risk associated with dietary exposure to iAs. Brazil is committed to the Sustainable Development Goals of the United Nations, one of which, End Hunger, also calls for food safety and sustainable agriculture [87]. Investments in research are essential to assess best practices for reducing iAs concentrations in rice, and public policies could provide support to rice producers through education and by promoting the adoption of good agriculture practices.

Pedron et al. [88] evaluated polishing and washing rice for potential implementation in the food industry. Polishing the grain removed 13–54% of total arsenic, depending on the duration of the polishing, which ranged from 20 s to 60 s. In the case of brown rice, washing the grains removed approximately 38.8% of total arsenic. Jo and Todorov [59] found that polishing brown rice can remove 16–33% of iAs. However, these practices are known to reduce some nutrients, as reported by Pedron et al. [88], who found that washing and polishing rice reduced the concentrations of nutrients (manganese, iron, cobalt, copper, zinc, and selenium) by 33–95%.

Another potential arsenic mitigation strategy is using fungi from the rhizosphere of rice. Segura et al. [89] tested two genera of fungi and obtained promising results. The authors concluded that direct application of *Aspergillus* sp. in soils might be a good alternative for reducing iAs concentration in rice grains.

The Joint FAO/WHO Code of Practice for the Prevention and Reduction of Arsenic Contamination in Rice states that national authorities should consider the implementation of measures directed at the sources of iAs and the adoption of specific agricultural practices. Authorities could determine which measures are most appropriate for their countries. Such measures include the identification and avoidance of arsenic sources: water for irrigation, contaminated soil, atmospheric emissions and wastewater from industry, materials used in agricultural and livestock production (pesticides, veterinary medicines, feed, soil amendments, and fertilizers), and waste from other materials (e.g., timber treated with copper chrome arsenate). Specific agricultural measures include education programs for farmers and implementing aerobic conditions or intermittent flooding during rice production, although only if cadmium concentrations in rice are not a concern [5].

There is a need for additional studies aimed at determining which mitigation strategies are the most suitable, taking into consideration the complexity of aspects related to agriculture, daycare centers, schools, maternity leave, and culture. The same interventions proposed in the risk assessment conducted by the U.S. FDA produced results that were less significant, possibly because rice consumption is lower in the United States than in Brazil, or because the dose–response model was different. In their risk assessment of exposure to iAs in rice [4], eliminating rice from the diet of infants and children ≤ 6 years of age would reduce the ILCR by 6%, imposing an MCL of 100 ng g^−1^ would reduce the ILCR by 4.3–18.3%, imposing an MCL of 75 ng g^−1^ would reduce the ILCR by 20–37%, and imposing an MCL of 50 ng g^−1^ would reduce the ILCR by 44.5%.

## 5. Conclusions

An important finding is that the ILCR, as well as the cancer risk for each age or age group, associated with exposure to white rice in Brazil is high, even when the iAs concentration is under the MCLs proposed by the FAO/WHO and BNMH. That might be attributed to the high level of rice consumption in the country, and the MCL established by the BNMH does not seem to be appropriate in view of the exposure scenario. The incremental cancer risk is highest at 35–65 years of age, when rice consumption is high given the lower body weight at that age, resulting in a higher dose of iAs and, consequently, a higher incremental cancer risk. The results are influenced by the exposure parameters adopted and possibly by some uncertainties related to them. A more extensive exposure assessment is needed. 

According to our findings, in Brazil, the risk associated with the consumption of brown rice appears to be lower than that associated with the consumption of white rice, given that we found the iAs concentration to be lower in brown rice. It is possible that the brown rice studied could be polished and sold as white rice. Thus, we can only conclude that samples of rice from some farms presented a lower concentration of iAs, and it is not specific to brown rice. Further studies are needed to verify our finding that some farms are producing rice with a lower concentration of iAs and, if verified, to implement interventions based on that understanding. We found some evidence that this low concentration of iAs could be explained by the variety of rice and by the practices adopted in rice fields.

The non-cancer risk associated with exposure to rice in Brazil is not concerning. 

The actual ILCR is probably higher than that found in this study, because we did not consider the presence of rice in other foods, such as infant cereal and formula. On the other hand, this probabilistic analysis has inherent limitations and is an indirect measure of risk. In addition, we presented some potentially efficient options for mitigating the risk, each of which could have social, political, and economic effects. Those effects should be evaluated in future studies.

## Figures and Tables

**Figure 1 ijerph-19-16460-f001:**
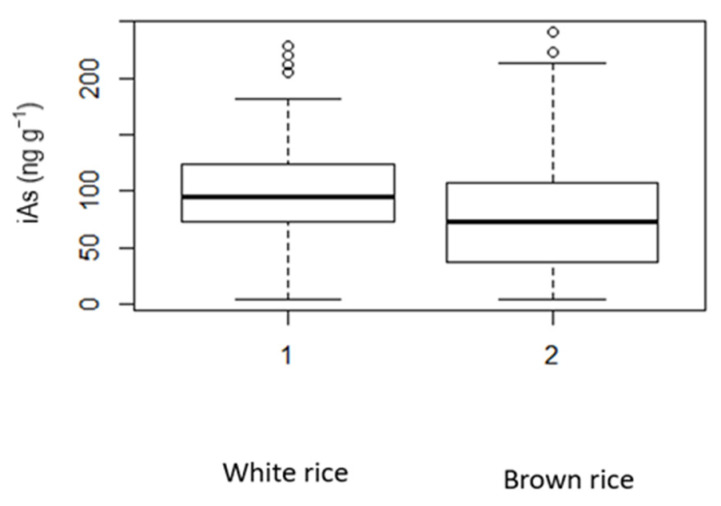
Box plot of inorganic arsenic (iAs) concentrations in polished (white) and husked (brown) rice.

**Figure 2 ijerph-19-16460-f002:**
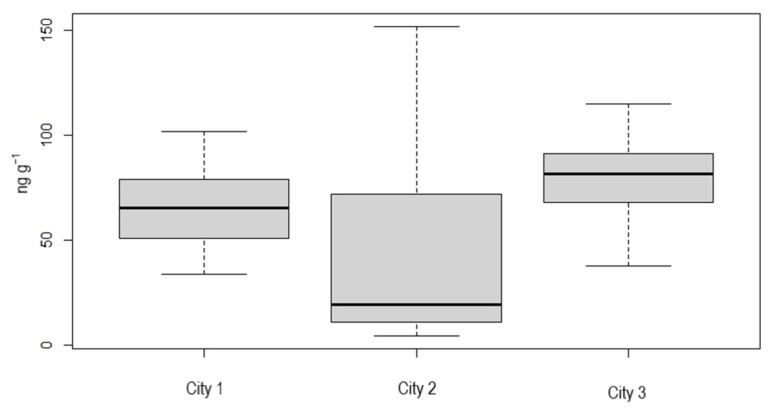
Box plot of inorganic arsenic (iAs) concentrations in brown rice from farms by city.

**Figure 3 ijerph-19-16460-f003:**
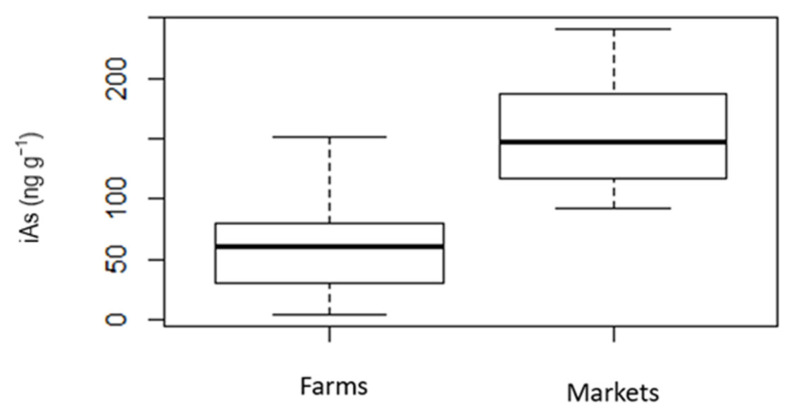
Box plot of inorganic arsenic (iAs) concentrations in husked (brown) rice from farms and markets.

**Figure 4 ijerph-19-16460-f004:**
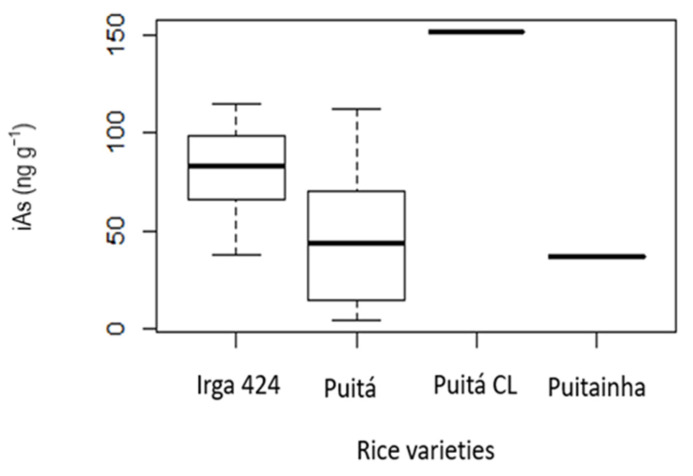
Box plot of inorganic arsenic (iAs) concentrations in four rice varieties.

**Table 1 ijerph-19-16460-t001:** Estimated incremental cancer risks and incremental lifetime cancer risk (means and 95th percentiles) associated with exposure to inorganic arsenic in polished (white) and husked (brown) rice.

Age (Years)	White Rice Cancer Risk and ILCR		Brown Rice Cancer Risk and ILCR	
	Mean	95th Percentile	Mean	95th Percentile
<1	1.62 × 10^−6^	2.79 × 10^−6^	6.47 × 10^−8^	9.08 × 10^−8^
1	9.19 × 10^−6^	1.58 × 10^−05^	3.66 × 10^−7^	5.14 × 10^−7^
2	7.39 × 10^−6^	1.27 × 10^−05^	2.95 × 10^−7^	4.14 × 10^−7^
3	7.23 × 10^−6^	1.24 × 10^−05^	2.88 × 10^−7^	4.05 × 10^−7^
4	7.02 × 10^−6^	1.21 × 10^−05^	2.80 × 10^−7^	3.93 × 10^−7^
5	5.52 × 10^−6^	9.48 × 10^−6^	2.20 × 10^−7^	3.09 × 10^−7^
6	5.09 × 10^−6^	8.75 × 10^−6^	2.03 × 10^−7^	2.85 × 10^−7^
7	4.52 × 10^−6^	7.76 × 10^−6^	1.80 × 10^−7^	2.53 × 10^−7^
8	4.07 × 10^−6^	7.00 × 10^−6^	1.62 × 10^−7^	2.28 × 10^−7^
9	3.56 × 10^−6^	6.12 × 10^−6^	1.42 × 10^−7^	1.99 × 10^−7^
10	3.12 × 10^−6^	5.36 × 10^−6^	1.24 × 10^−7^	1.74 × 10^−7^
11	2.77 × 10^−6^	4.75 × 10^−6^	1.10 × 10^−7^	1.55 × 10^−7^
12	2.45 × 10^−6^	4.21 × 10^−6^	9.76 × 10^−08^	1.37 × 10^−7^
13	2.21 × 10^−6^	3.80 × 10^−6^	8.82 × 10^−08^	1.24 × 10^−7^
14	2.06 × 10^−6^	3.54 × 10^−6^	8.21 × 10^−08^	1.15 × 10^−7^
15	1.92 × 10^−6^	3.30 × 10^−6^	7.66 × 10^−08^	1.07 × 10^−7^
16	1.85 × 10^−6^	3.19 × 10^−6^	7.40 × 10^−08^	1.04 × 10^−7^
17	1.79 × 10^−6^	3.08 × 10^−6^	7.15 × 10^−08^	1.00 × 10^−7^
18	1.75 × 10^−6^	3.00 × 10^−6^	6.96 × 10^−08^	9.77 × 10^−08^
19	1.79 × 10^−6^	3.07 × 10^−6^	7.13 × 10^−08^	1.00 × 10^−7^
20 to <25	8.58 × 10^−6^	1.47 × 10^−05^	3.42 × 10^−7^	4.80 × 10^−7^
25 to <30	8.20 × 10^−6^	1.41 × 10^−05^	3.27 × 10^−7^	4.59 × 10^−7^
30 to <35	8.05 × 10^−6^	1.38 × 10^−05^	3.21 × 10^−7^	4.50 × 10^−7^
35 to <45	1.59 × 10^−05^	2.73 × 10^−05^	6.32 × 10^−7^	8.88 × 10^−7^
45 to <55	1.57 × 10^−05^	2.70 × 10^−05^	6.27 × 10^−7^	8.80 × 10^−7^
55 to <65	1.46 × 10^−05^	2.51 × 10^−05^	5.82 × 10^−7^	8.18 × 10^−7^
65 to <70	3.46 × 10^−6^	5.94 × 10^−6^	1.38 × 10^−7^	1.94 × 10^−7^
ILCR	1.51 × 10^−4^	2.60 × 10^−4^	6.04 × 10^−6^	8.47 × 10^−6^

ILCR: incremental lifetime cancer risk.

**Table 2 ijerph-19-16460-t002:** Estimated fractional hazard quotients by age and hazard quotient for a lifetime for exposure to inorganic arsenic in polished (white) and husked (brown) rice.

Age (Years)	White Rice		Brown Rice	
	Hazard Quotient		Hazard Quotient	
	Mean	95th Percentile	Mean	95th Percentile
<1	3.61 × 10^−3^	6.20 × 10^−3^	1.44 × 10^−4^	2.04 × 10^−4^
1	2.04 × 10^−2^	3.51 × 10^−2^	8.15 × 10^−4^	1.15 × 10^−3^
2	1.64 × 10^−2^	2.82 × 10^−2^	6.55 × 10^−4^	9.27 × 10^−4^
3	1.61 × 10^−2^	2.76 × 10^−2^	6.41 × 10^−4^	9.07 × 10^−4^
4	1.56 × 10^−2^	2.68 × 10^−2^	6.23 × 10^−4^	8.81 × 10^−4^
5	1.23 × 10^−2^	2.11 × 10^−2^	4.89 × 10^−4^	6.92 × 10^−4^
6	1.13 × 10^−2^	1.94 × 10^−2^	4.51 × 10^−4^	6.38 × 10^−4^
7	1.00 × 10^−2^	1.73 × 10^−2^	4.00 × 10^−4^	5.67 × 10^−4^
8	9.06 × 10^−3^	1.56 × 10^−2^	3.61 × 10^−4^	5.11 × 10^−4^
9	7.93 × 10^−3^	1.36 × 10^−2^	3.16 × 10^−4^	4.47 × 10^−4^
10	6.93 × 10^−3^	1.19 × 10^−2^	2.76 × 10^−4^	3.91 × 10^−4^
11	6.15 × 10^−3^	1.06 × 10^−2^	2.45 × 10^−4^	3.47 × 10^−4^
12	5.45 × 10^−3^	9.35 × 10^−3^	2.17 × 10^−4^	3.07 × 10^−4^
13	4.92 × 10^−3^	8.45 × 10^−3^	1.96 × 10^−4^	2.77 × 10^−4^
14	4.58 × 10^−3^	7.86 × 10^−3^	1.82 × 10^−4^	2.58 × 10^−4^
15	4.27 × 10^−3^	7.34 × 10^−3^	1.70 × 10^−4^	2.41 × 10^−4^
16	4.13 × 10^−3^	7.09 × 10^−3^	1.64 × 10^−4^	2.33 × 10^−4^
17	3.99 × 10^−3^	6.85 × 10^−3^	1.59 × 10^−4^	2.25 × 10^−4^
18	3.88 × 10^−3^	6.67 × 10^−3^	1.55 × 10^−4^	2.19 × 10^−4^
19	3.98 × 10^−3^	6.83 × 10^−3^	1.59 × 10^−4^	2.24 × 10^−4^
20 to <25	1.91 × 10^−2^	3.28 × 10^−2^	7.60 × 10^−4^	1.08 × 10^−3^
25 to <30	1.82 × 10^−2^	3.13 × 10^−2^	7.27 × 10^−4^	1.03 × 10^−3^
30 to <35	1.79 × 10^−2^	3.07 × 10^−2^	7.14 × 10^−4^	1.01 × 10^−3^
35 to <45	3.53 × 10^−2^	6.06 × 10^−2^	1.41 × 10^−3^	1.99 × 10^−3^
45 to <55	3.50 × 10^−2^	6.01 × 10^−2^	1.39 × 10^−3^	1.97 × 10^−3^
55 to <65	3.25 × 10^−2^	5.58 × 10^−2^	1.30 × 10^−3^	1.83 × 10^−3^
65 to <75	7.69 × 10^−3^	1.32 × 10^−2^	3.07 × 10^−4^	4.34 × 10^−4^
0 to <70	3.37 × 10^−01^	5.78 × 10^−01^	1.34 × 10^−2^	1.90 × 10^−2^

**Table 3 ijerph-19-16460-t003:** Estimated incremental lifetime cancer risk and parameters for hypothetical scenarios of interventions to reduce the cancer risk of exposure to inorganic arsenic in Brazil.

Scenario	iAs Limit	N	iAs Concentration	Distribution (Mean; SD or μ; δ)	ILCR	Reduction in ILCR	
			(ng g^−1^)				
	(ng g^−1^)		Mean; SD		Mean	95th Percentile	(%)
1. Consumption of brown rice from farms with low iAs levels	151.9	69	57.36; 34.36	Log-normal (3.77; 0.87)	5.73 × 10^−6^	8.13 × 10^−6^	5.03
2. Consumption of brown rice of the Puitá variety	112	48	46.58; 31.68	Log-normal (3.51; 0.91)	5.38 × 10^−6^	7.95 × 10^−6^	10.85
3. Imposition of a white rice MCL of 100 ng g^−1^	100	37	71.94; 22.47	Normal (71.94; 22.47)	1.10 × 10^−4^	1.66 × 10^−4^	27.29
4. Imposition of a white rice MCL of 75 ng g^−1^)	75	17	53.74; 21.15	Normal (53.74; 21.15)	8.27 × 10^−05^	1.35 × 10^−4^	45.39
5. Imposition of a white rice MCL of 50 ng g^−1^)	50	6	30.55; 16.85	Normal (30.55; 16.85)	4.88 × 10^−05^	8.95 × 10^−05^	67.79
6. No consumption of white rice ≤ 6 years of age	200	64	100.17; 44.62	Normal (100.17; 44.62)	1.17 × 10^−4^	1.99 × 10^−4^	22.67

iAs: inorganic arsenic; ILCR: incremental lifetime cancer risk; MCL: maximum contaminant level.

## Data Availability

The data presented in this study are available in the Appendix A.

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
