# Peer review of "Exposure to Inorganic Arsenic in Rice in Brazil: A Human Health Risk Assessment"

_ijerph, 2022, doi:10.3390/ijerph192416460_

Round 1

Reviewer 1 Report

This is probability-based and ecologic analysis of arsenic exposure in rice and potential cancer and non-cancer risk outcomes.

The topic if arsenic and cancer is not novel, but the hypothesis is ongoing. This study only considered overall cancer risk, not specific cancers, which is a limitation. The epidemiologic approach has inherent limitations, which should be mentioned in the paper. The authors could address what the strength and novelty of their analysis.

The biggest positive about this analysis is that it provides information about a arsenic and outcomes risk relevant to Brazil, and the specific cities evaluated. It doesn't add to the understanding about the etiology of arsenic and cancer risk.The methodology has inherent limitations, as does any study. This is not a population based analysis, and therefore does not provide odds of risk. The cancer risk is derived from the slope estimate; potential confounders are not considered. I suggest consulting with a statistician to evaluate the statistical methods.

The conclusions consistent with the evidence and arguments presented and address the main question posed, but I do think that the wording could be improved. It is easy to misunderstand the authors use of 'cancer risk' as a odds ratio, which it is not. They may be over-stating the interpretation of their statistical findings.

About the references, I suggest consulting someone who has published on arsenic exposure and cancer.

The tables and figures could add footnotes for clarity.

Reviewer 2 Report

In this article, cancer risk associated with exposure to inorganic arsenic in rice in Brazil has been assessed. I would accept this article after minor revisions by taking into account these recommendations.

QUESTIONS AND COMMENTS:

In general terms, authors think it would have been more appropriate to compare white and brown rice from the same location. How valid are your conclusions based on this fact? In this case, article writing might be reconsidered 

-In line 10, I guess there is an “and” missing between “cancer non-cancer effects” ¿?

- In lines 44-45, it would be interesting to indicate the As MCL value and the reference of FAO/WHO organisations where this information can be found.

- In lines 47-48 you say “The average daily consumption of rice by adults is estimated to be 167 g”. Could the authors specify which organization or study establishes this value?

- In line 60, author say: “Only a few studies have evaluated exposure to iAs in rice in Brazil”. Please, read and consider including (if necessary) the following references related to As content in Brazilian rice:

1. https://doi.org/10.1016/j.jfca.2021.103853

2. https://doi.org/10.1016/j.jfca.2021.103914

3. https://doi.org/10.1016/j.ecoenv.2018.11.025

4. https://doi.org/10.1016/j.foodchem.2019.02.043

5. https://doi.org/10.1016/j.foodres.2016.07.011

- Revise the tables and write all the superscripts correctly

- in section “2.1 Average daily dose, cancer risk, and hazard quotient”  there should appear the meaning of the values for ADD, ILCR and HQ. For example, “An HQ above 1 indicates that the dose ingested is higher than the reference dose and that there is a potential for adverse effects, in this context including cardiovascular and dermal effects” should be in this section. Same for the rest.

- in Table 1, first column is age, the word “ILCR” should appear in the other columns.

Reviewer 3 Report

In this manuscript, the authors assessed the cancer and non-cancer risks of Brazilian populations exposure to inorganic arsenic (iAs) in white and brown rice was reviewed according to a study of Bastista (2015). Meanwhile, some mitigation measures are suggested to be reduce the risk of exposure to iAs due to rice consumption. The study is important to understand and reduce the exposure risk of iAs in rice. Specific comments are as follows:

1. In abstract, cancer non-cancer effects should be cancer and non-cancer effects.

2. Keywords are inappropriate, how about inorganic arsenic, rice, dietary exposure risk, mitigation measure, and Brazil?

3. In part 2.2, the authors introduce that the total concentration of As in rice were analyzed by ICP-MS and the samples for speciation analysis were analyzed by HPLC-ICP-MS. In the speciation analysis, did the obtained results represent iAs or organic As concentration? If it is organic As, the concentration of iAs should be total As minus organic As. Please express clearly.

4. line 125-128, the authors state that ……in which 24-h duplicate diet samples were analyzed by inductively coupled plasma mass spectrometry…..; generally, ICP-MS are used to analyze the concentration of elements, here, how to obtain the consumption data of rice?

5. line 179, The incremental risk is higher for infants and children 1 6 years of age. This sentence should be revised.

6. Line 191, 1.34 x10-2.for brown rice, the dot between 1.34x10-2 and for should be removed.

7. Line 198, what dose BNMH mean?

8. Line 311-312, Because the standard maternity leave in Brazil is 120 daysb[45], please check the unit.

9. Line 438, f before from should be removed.

Round 2

Reviewer 2 Report

Dear authors, 

I do not see the inclusion of the suggested references (not track in red). Please, check it.

 In line 60, author say: “Only a few studies have evaluated exposure to iAs in rice in Brazil”. Please, read and consider including (if necessary) the following references related to As content in Brazilian rice:

1. https://doi.org/10.1016/j.jfca.2021.103853

2. https://doi.org/10.1016/j.jfca.2021.103914

3. https://doi.org/10.1016/j.ecoenv.2018.11.025

4. https://doi.org/10.1016/j.foodchem.2019.02.043

5. https://doi.org/10.1016/j.foodres.2016.07.011

Author Response

Dear reviewer,

We have added the suggested references.

Thank you.